# Negative feedback may suppress variation to improve collective foraging performance

**Andreagiovanni Reina**[1,2]\*, **James A. R. Marshall**[1,3]\*

**1** Department of Computer Science, University of Sheffield, United Kingdom, **2** IRIDIA, Université Libre de Bruxelles, Belgium, **3** Opteran Technologies Limited, Sheffield, United Kingdom

\* andreagiovanni.reina@ulb.be (AR); james.marshall@sheffield.ac.uk (JARM)

**Data Availability Statement:** All data, simulation code, and a Mathematica notebook reproducing the analyses presented herein are available in the GitHub repository https://github.com/

## Abstract

Social insect colonies use negative as well as positive feedback signals to regulate foraging behaviour. In ants and bees individual foragers have been observed to use negative phero-mones or mechano-auditory signals to indicate that forage sources are not ideal, for example being unrewarded, crowded, or dangerous. Here we propose an additional function for negative feedback signals during foraging, variance reduction. We show that while on aver-age populations will converge to desired distributions over forage patches both with and without negative feedback signals, in small populations negative feedback reduces variation around the target distribution compared to the use of positive feedback alone. Our results are independent of the nature of the target distribution, providing it can be achieved by forag-ers collecting only local information. Since robustness is a key aim for biological systems, and deviation from target foraging distributions may be costly, we argue that this could be a further important and hitherto overlooked reason that negative feedback signals are used by foraging social insects.

## Author Summary

Social insect colonies regulate the number of insects foraging at different food sources through a combination of positive and negative feedback signals. Through positive feed-back signals—such as ants' pheromone trails and bees' waggle dances—insects recruit each other to increase the number of foragers committed to a food source that has been evaluated as profitable. Negative feedbacks are instead inhibitory signals that are delivered to reduce commitment to a food source where an unfavourable change has been detected, for example the arrival of a predator or a decrease in nutritional reward. Our mathemati-cal analysis explains an additional function for negative feedback; inhibitory signals can also be useful in static conditions to reduce the variance in the number of insects allocated to each food source, thus better distributing insects among the available sources. Our results can help explain field observations that are not fully understood yet, such as the periodic delivery of a small number of inhibitory signals among honeybees visiting the same forage patch even in static conditions.

DiODeProject/VarianceSuppression. This is supplemented by a notebook for MuMoT, an open-source tool for multiscale modelling, which reproduces similar results to those presented herein. The notebook is available online at https://mybinder.org/v2/gh/DiODeProject/MuMoT/master?filepath=DemoNotebooks%2FVariance_suppression.ipynb and https://github.com/DiODeProject/MuMoT/blob/master/DemoNotebooks/Variance_suppression.ipynb.

**Funding:** A.R. acknowledges support from the Belgian F.R.S.-FNRS, of which he is a Chargé de Recherches (https://www.frs-fnrs.be). J.A.R.M. was funded by the European Research Council (ERC) under the European Union's Horizon 2020 research and innovation programme (grant agreement number 647704), https://erc.europa.eu/funding/. The funders had no role in study design, data collection and analysis, decision to publish, or preparation of the manuscript.

**Competing interests:** The authors have declared that no competing interests exist.

## Introduction

Collectively-foraging social insects use feedback mechanisms in order to robustly and efficiently satisfy the nutritional requirements of the colony. Positive feedback signal usage by such foraging social insects is well known, such as mass-recruitment via pheromone in various ant species [1], and recruitment of small numbers of individuals such as via the honeybees' waggle-dance [2], or rock ants' tandem-running [3]. The use of negative feedback signals in these systems has, however received comparatively little attention. Negative feedback was predicted to be important for collectively foraging species [4, 5], and subsequently discovered in diverse systems such as Pharaoh's ants [6, 7] and honeybees [8, 9]. Several studies have interpreted negative feedback as a mechanism to reduce recruitment to a resource based on some aspect of its quality, for example allowing unrewarded trails to be shut down [6, 7], allowing recruitment to a crowded source of forage to be reduced [10], or transferring information that a forage patch may have an increased predation risk [8, 11]. Subsequent studies have similarly focussed on the role of negative feedback in dealing with time-varying forage patches [12, 13], or with the amount of available comb storage space [14].

Here we propose an alternative function for negative feedback mechanisms in collective foraging, suppression of costly variation in the colony's foraging performance. In the following, we present simple models of collective foraging with positive and negative feedback, and with positive feedback only. We show how both models are able to approach a desired target distribution over forage patches on average, when forager populations are assumed to be infinite. However, when finite forager populations are modelled, the two foraging systems differ in the robustness with which they achieve the target distribution; with positive feedback only, stochastic fluctuations can lead to the forager population being far from its target distribution at any point in time, however by adding negative feedback the forager distribution becomes more robust. We argue that this will increase colony-level foraging success [15, 16], and thus may represent a new functional explanation for the observation of negative feedback in foraging by social insect colonies.

Foraging theory is an active and complex research area, and our results do not rely on assumptions about the nature of the colony's target distribution, other than it can be achieved by agents with access only to local information at both the forage source, and the colony. Thus, the target distribution may be akin to an Ideal Free Distribution, in which agents are distributed such that none can improve overall foraging efficiency by switching to a different forage patch [16, 17]. Alternatively, the target distribution may be based on the requirement of the colony for different micro- and macro-nutrients [18–21]. Or, the target distribution may be based on some other objective entirely, or on combinations of objectives such as those just discussed. In ignoring the nature of the distribution, therefore, our focus is purely on the dynamics of foraging, and how negative feedback can improve this.

For our analysis we adapt our model from a simple model of negative feedback for foraging in honeybee colonies [12], in itself inspired by models of negative feedback in house-hunting honeybee swarms [22–24]; however since other social insect species such as Pharaoh's Ants also make use of negative feedback during foraging [6], we argue that the model is generally applicable.

## Methods

We assume a target distribution of the individuals to the $n$ patches in quantities proportional to the relative patch quality arbitrarily defined:

$$x_i^* \approx \frac{q_i}{\sum_{j \in n} q_j}, \qquad i \in \{1, \ldots, n\}, \tag{1}$$

**Table 1. The two analysed models can be described in terms of transitions between commitment states by individuals.** The commitment states are 'committed to foraging patch $i$' ($X_i$) or 'uncommitted' ($X_U$). Both models have the same positive and negative feedback for *independent transitions*: quality-dependent discovery and constant abandonment (leak $a$). The difference lies in the social feedback; one model (blue) has quality-insensitive recruitment ($r_i = \rho$) but no negative social feedback ($z = 0$). The other model (red) has both quality-sensitive recruitment ($r_i = \rho q_i$) and quality-insensitive self-inhibition ($z > 0$), as reported by field observations [30]. In these representative models, we set rates as constant and (linear) quality-sensitive functions of the quality according to the best function we obtain with numerical optimisation (see S1 Text).

| | | WITHOUT NEGATIVE SOCIAL FEEDBACK | WITH NEGATIVE SOCIAL FEEDBACK |
|---|---|---|---|
| Independent discovery | $X_U \xrightarrow{q_i} X_i$ | Quality-sensitive | Quality-sensitive |
| Independent abandonment (leak) | $X_i \xrightarrow{a} X_U$ | Constant | Constant |
| Recruitment (positive social feedback) | $X_U + X_i \xrightarrow{r_i} 2X_i$ | Constant | Quality-sensitive |
| Stop signalling (negative social feedback) | $X_i + X_i \xrightarrow{z} X_U + X_i$ | | Constant |

where $q_i$ is the quality of patch $i$. In our models an individual's state can be either uncommitted ($X_U$) or committed to patch $i$ ($X_i$) with $i \in \{1, \ldots, n\}$. Therefore, based on the number of patches $n$, the commitment of the population will be split among $n + 1$ subpopulations; we represent the subpopulation proportions as $x_U$ and $x_i$, in the closed interval [0, 1]. Note that, in a finite population of $S$ individuals, it will be impossible for the colony to achieve exactly the desired target distribution if $x_i S$ is not an integer number.

We analyse the population dynamics of the two systems parametrised to reach the same target distribution (with and without negative social feedback) using mean-field models of infinite and finite populations, using ordinary differential equations (ODEs) and stochastic simulation of the master equation respectively. Both types of analyses can be performed for models derived from chemical reaction equations, which specify how individuals in the system interact and change state (see Table 1).

The ODE model assumes an infinitely-large population size $S$ and provides deterministic system dynamics in the absence of any noise from finite population effects. On the other hand, stochastic simulation of the master equation (Gillespie's SSA [25]) gives a probabilistically correct simulation of dynamics of finite populations of size $S$.

While previous research has documented that collective foraging is regulated by the actions and interactions that we included in our models, the relationship between their frequency (transition rates) and the estimated nest-site quality are still debated. Table 1 reports the best functions we obtained through numerical optimisation to approximate the target distribution. Including negative feedback inevitably requires a change also in the recruitment function, from constant to linearly proportional to the quality. The difference in the recruitment function between the two models has not been imposed by design but is the result of the numerical optimisation analysis that we show in detail in S1 Text. Here, we assume that social recruitment (positive feedback) is much more efficient than independent discovery, so $r_i \gg q_i$, as has been documented in a large variety of social insect species [26–29]. For fair comparison, the average recruitment strength $r$ is equalised between the two models so that quality-sensitive recruitment transitions—model with negative feedback—happens on average at the same rate of quality-insensitive recruitment—model without negative feedback (see S2 Text). The model with only positive feedback is easy to solve for the desired equilibrium distribution of foragers, with a simple parameterisation of individuals' rates (see S3 Text). The model with negative feedback, however, requires a heuristic individual parameterisation based on site qualities, which we perform numerically. However, this heuristic has a simple functional form (see Fig A in S2 Text) so could easily be approximated by real foragers.

## Results

The two top panels of Fig 1 show the time dynamics of the two models for representative values and $n = 3$ patches. Both models asymptotically approximate the target distribution of Eq (1).

Through numerical integration of the master equations, we investigate the effect of stochastic fluctuations on the system dynamics [25]. The fluctuation size is inversely proportional to the system size $S$, *i.e.* there are no fluctuations in very large groups (*i.e.* $S \to \infty$) and large

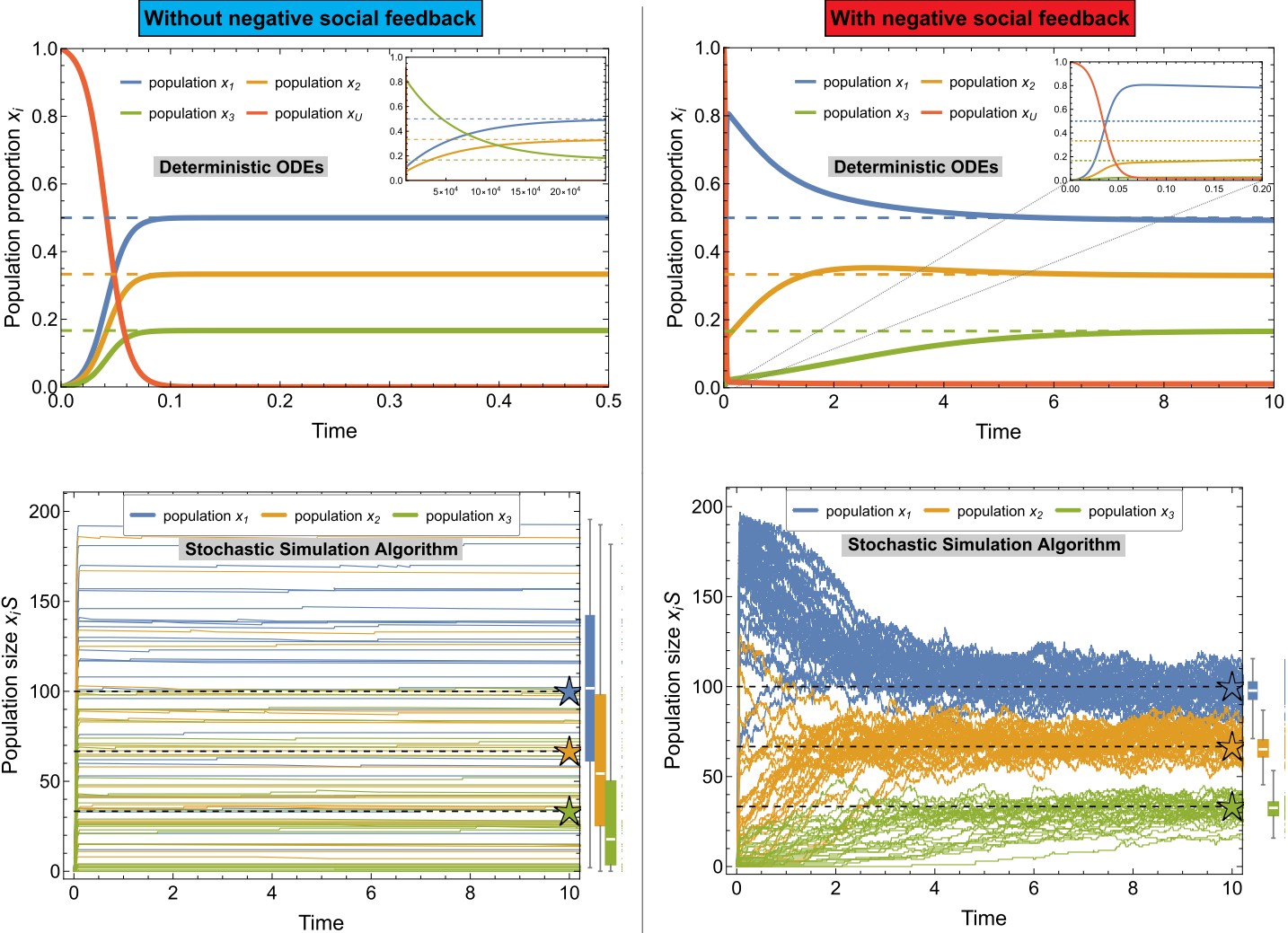

**Fig 1. Temporal evolution of the models without (left) and with (right) negative social feedback in an environment with $n = 3$ food patches with qualities $q_1 = 0.75$, $q_2 = 0.5$, $q_3 = 0.25$.** The top panels show the dynamics of the ODEs for systems of infinite size $S \to \infty$. The bottom panels show the trajectories of 30 representative runs of the stochastic simulation algorithm (SSA, [25]) for a system comprised of $S = 200$ individuals. The boxplots on the right of each bottom panel show the statistical aggregate at time 400 for 1000 runs of the SSA. (Other simulation parameters are: constant abandonment $a = 10^{-3}$, average recruitment strength $r = 100$, and stop signal strength $z \simeq 3.1$.) While the infinite size dynamics predict convergence to the target distribution of Eq (1) (dashed lines) for both models, the stochastic trajectories show different results for the two models. The system without negative social feedback has smaller fluctuations over time but frequently stabilises at values far from the target distribution (bottom-left panel). The system with negative social feedback fluctuates more but always remains relatively close to the target distribution (bottom-right panel). The apparently quicker dynamics of the ODE model for the system without negative social feedback are due to the symmetric initial conditions. In the left inset, we show that a small perturbation of the initial population (*i.e.* $x_1$, $x_2 = 0$ and $x_3 = 0.05$) delays the convergence by more than 5 orders of magnitude. Such a susceptibility to random fluctuations is made evident by the stochastic trajectories. The right inset shows a zoom of the larger plot.

fluctuation in small groups. The effect of the system-size noise can be appreciated in the two bottom panels of Fig 1. They show 30 representative trajectories for a system of size $S = 200$. The higher variance can also be appreciated in the boxplots on the right of each bottom panel of Fig 1, in which the average of 1,000 simulations hits the target value in both models; however, the variance is reduced considerably with the introduction of negative social feedback. These results are not specific to the representative example of Fig 1, but are consistent throughout the wide parameter space (see analysis in S4 Text). Additionally, increasing abandonment, which is a form of independent, asocial negative feedback, is not sufficient to reduce variance (see S5 Text).

Large deviations from the target distribution could compromise the ability of the colony to intake the necessary nutrients for survival and reproduction, thus decreasing colony fitness. Fig 2 shows how the error in achieving the target distribution is significantly higher without negative social feedback. Similarly, the speed of adaptation to environmental changes is an important factor in the survival of the colony [31, 32]. The system without negative feedback can be incapable of adapting to changes in a timely manner because its temporal dynamics vary significantly depending on the initial commitment (see top-left inset of Fig 1). The system with negative feedback, instead, displays a constant convergence time regardless of the initial state of the system (see S6 Text). Fig 3 shows how the convergence speed and the deviation from the target distribution are influenced by the strength of the negative feedback; the strength of negative feedback can tune a speed-robustness trade-off, similarly to the tuning of speed-value and speed-coherence trade-offs in consensus decisions [23, 24, 33]. In agreement with field observations of honeybees, which increase stop signalling when a quick response is necessary [10], our analysis also predicts a speed-up of the group dynamics for higher levels of

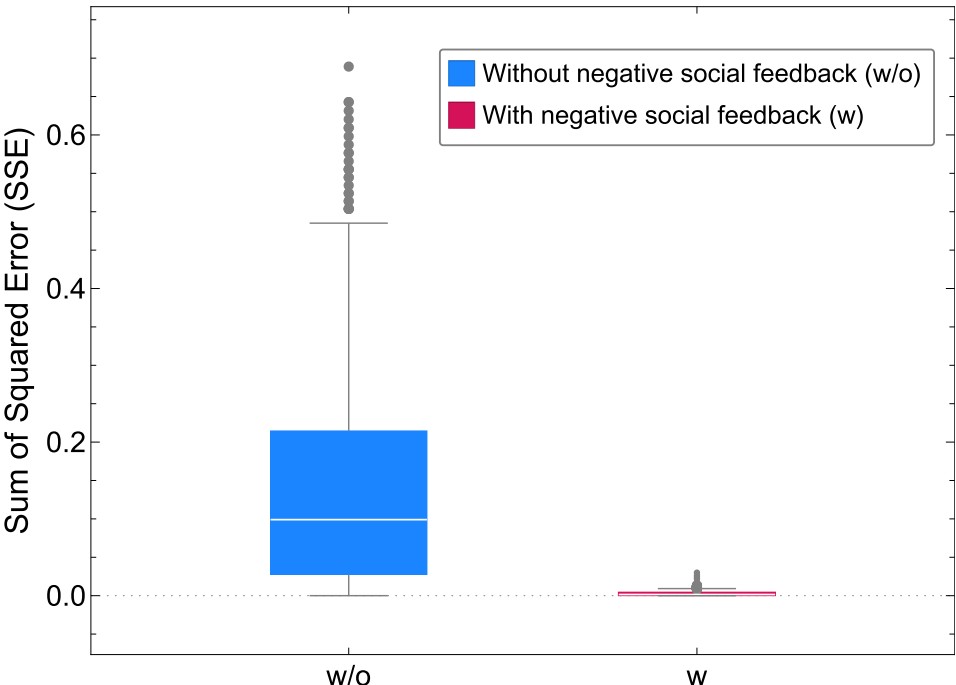

**Fig 2. Sum of squared errors (SSE) computed as the sum for $n = 2$ food patches of the square of the difference between the subpopulation size at time 1000 (convergence) and the target distribution to that patch (see S7 Text).** The boxplots show the distribution of the SSE for $10^3$ numerical simulations for swarm size $S = 200$, average recruitment strength $r = 100$, and qualities $q_1 = 0.75$ and $q_2 = 0.5$.

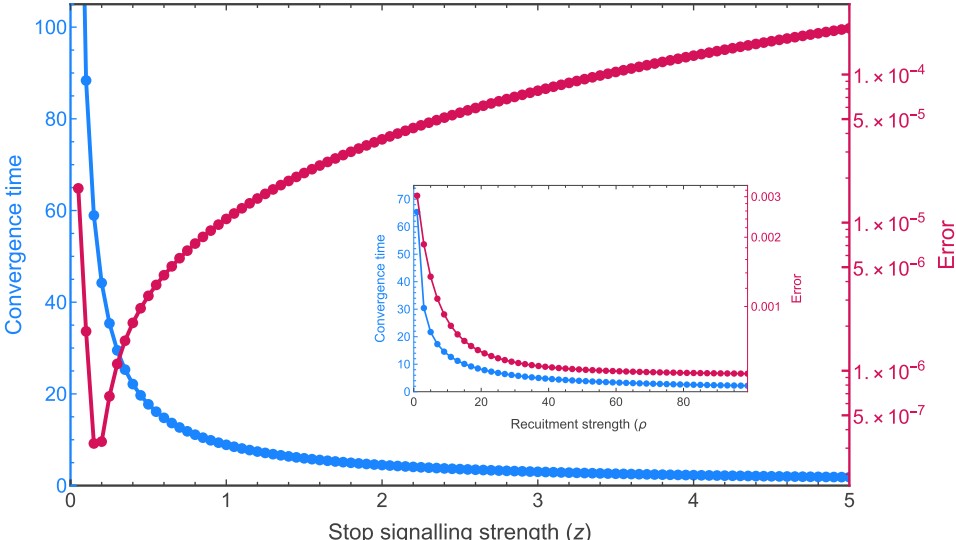

**Fig 3. The stop signalling strength can be the control parameter in a speed-robustness trade-off.** Stronger stop signalling speeds up the convergence of the system (magenta curve) but also increases the predicted error from the target distribution (blue curve). These results are in agreement with field observations that documented an increase in stop signalling when a quick response to environmental changes was necessary [10]. The inset shows that for increasing recruitment strength $\rho$ both convergence time and error decrease. Both error and convergence time are computed from the infinite population model (ODE). The error is computed as the sum for every foraging population of the squared distance $R^2$ from the target at large time (convergence, computed analytically as the ODE's stable fixed point in the unit-simplex). The convergence time (magenta curve) is computed as the time necessary to reach the (numerically computed) fixed point. As the system has an asymptotic convergence, the reported time corresponds to the $R^2$ error becoming smaller than $10^{-4}$.

negative feedback. The dynamics of Fig 3 can also be discussed in light of the results of a recent empirical study which has found that sensitivity to inhibitory signalling changes as a function of colony size [34]. Our model, by linking the negative feedback rate with convergence speed and the deviation from the target distribution, offers predictions that can be tested on colonies of different sizes. For example, the time to adaptation to environmental changes is expected to be longer in smaller colonies, where bees show a lower sensitivity to inhibitory signalling [34], than in larger colonies.

## Discussion

Negative feedback has been considered in collective decisions, particularly as a means of symmetry breaking [22–24, 35], and in foraging, as a means of adapting to dynamically changing environments [7, 10, 12, 13]. Other than in entomology, negative feedback has been observed as a tool for noise reduction in gene networks [36–38] and in electronic systems [39, 40]. Here we have shown that negative feedback may play an important role in reducing variance in colony foraging performance. For example, considering the honeybee system that inspired our model, field observations have reported that levels of stop signalling increase in response to changes such as dangerous, overcrowded, or depleted food patches [10, 11, 13, 41]; however, it has not yet been fully understood why, even in static conditions, honeybees always deliver a small number of stop signals to foragers visiting the same forage patch [10, 13, 34]. This pattern is consistent with our model, and the analysis presented is an interpretation for such observed behaviour.

Our results suggest a further progression in the evolution of collective foraging behaviour; solitary foraging by members of social insect colonies evolved first, but was comparatively inefficient due to the need for foragers to repeatedly and independently discover forage sites [42, 43] (see S8 Text). Subsequently, positive social feedback evolved to improve foraging efficiency [44–46], but this came at the expense of robustness of the foraging outcome, through increased variance in foraging performance (see S9 Text). Finally, negative feedback evolved not only to respond better to changing environments, but also to reduce variance in foraging performance. The re-use of negative feedback signals, such as in the case of honeybee stop-signals which are used in both foraging [10, 34] and house-hunting [22] life history stages, would facilitate performance-enhancing innovations in signalling behaviours; however, it is not clear whether stop-signalling first arose in foraging or in house-hunting contexts (intuitively, we suggest the former, a more common life history event).

Some species have not evolved negative signalling mechanisms but rely on natural decay of feedback, such as pheromone evaporation. For instance, *Lasius niger* ants rely on the downregulation of positive feedback (*i.e.* pheromone deposition) in order to let pheromone decay take over [47, 48]. It is worth noting that this is not technically negative feedback; given the time taken from the first observations of the positive-feedback signals in colonies of honeybees and ants [3, 49, 50] to that of the corresponding negative feedback signals [6, 51], it may be worth further exploring social insects in which explicit negative feedback has not been observed, to search for expected negative feedback mechanisms, or explain why their life history means they would not be beneficial. As a motivating example, decreases in the number of waggle circuit repetitions in honeybee swarms were taken to be due to decay processes internal to scout bees [52], but the negative stop-signal was subsequently discovered to be significant in these swarms [22].

We conclude by noting that our study highlights the importance of using multiscale modelling to understand collective behaviour [53–55]. In fact, through mean-field analysis we could not observe the dynamics that justify the use of the negative feedback. Instead, complementing the analysis with probabilistic models, we have been able to identify the system dynamics that favour the appearance of stop signalling as a mechanism for variance reduction. Multiscale modelling is a valuable framework which combines the use of a set of modelling techniques to analyse the system at various levels of complexity and noise. In this study, we only employed noise-free mean-field analysis and master equations with system-size dependent noise. However, further analysis could include the impact of spatial noise, and time-correlated information and/or interactions [53].

## Supporting Information

**S1 Text. Relationship between transition rates and food patch quality.**
(PDF)

**S2 Text. Parameters of the models.**
(PDF)

**S3 Text. Stability analysis.**
(PDF)

**S4 Text. Variance reduction in a large parameter space.**
(PDF)

**S5 Text. Effect of asocial negative feedback (abandonment or leak $\alpha$).**
(PDF)

**S6 Text. Adapting to changing conditions.**
(PDF)

**S7 Text. Sum of squared error.**
(PDF)

**S8 Text. Asocial model.**
(PDF)

**S9 Text. Large deviation from target.**
(PDF)

# Acknowledgments

The authors thank Cristian Fernández-Oto, Gianni Di Caro, Ulrike Feudel, and David Wagg for useful discussions.

# Author Contributions

**Conceptualization:** Andreagiovanni Reina, James A. R. Marshall.

**Data curation:** Andreagiovanni Reina.

**Formal analysis:** Andreagiovanni Reina, James A. R. Marshall.

**Funding acquisition:** Andreagiovanni Reina, James A. R. Marshall.

**Investigation:** Andreagiovanni Reina, James A. R. Marshall.

**Methodology:** Andreagiovanni Reina.

**Project administration:** Andreagiovanni Reina.

**Software:** Andreagiovanni Reina.

**Visualization:** Andreagiovanni Reina.

**Writing – original draft:** Andreagiovanni Reina, James A. R. Marshall.

**Writing – review & editing:** Andreagiovanni Reina, James A. R. Marshall.

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
