## [Decision Letter · Decision Letter 0]

8 Mar 2022

Dear Dr Reina,

Thank you very much for submitting your manuscript "Negative feedback may suppress variation to improve collective foraging performance" for consideration at PLOS Computational Biology. As with all papers reviewed by the journal, your manuscript was reviewed by members of the editorial board and by several independent reviewers. The reviewers appreciated the attention to an important topic. Based on the reviews, we are likely to accept this manuscript for publication, providing that you modify the manuscript according to the review recommendations.

The reviewers all agree that the presented results are valuable and worth publishing. In addition, they make a number of constructive suggestions to further increase the clarity of the paper. In particular, they suggest to

(-) discuss, in the main text, the exact differences between the two models. In particular, it would be useful to have a detailed discussion why recruitment is labeled as constant in the model without negative social feedback, but labeled as quality-sensitive in the model with negative social feedback.

(-) provide more detail about the mechanism that drives the observed reduction in variance, and

(-) move some of the supplementary material to the main text.

I concur with all these suggestions, and I'd like to encourage the authors to make the corresponding changes.

Sincerely,

Christian Hilbe

Associate Editor

PLOS Computational Biology

Natalia Komarova

Deputy Editor

PLOS Computational Biology

[LINK]

The reviewers all agree that the presented results are valuable and worth publishing. In addition, they make a number of constructive suggestions to further increase the clarity of the paper. In particular, they suggest to

(-) discuss, in the main text, the exact differences between the two models. In particular, it would be useful to have a detailed discussion why recruitment is labeled as constant in the model without negative social feedback, but labeled as quality-sensitive in the model with negative social feedback.

(-) provide more detail about the mechanism that drives the observed reduction in variance, and

(-) move some of the supplementary material to the main text.

I concur with all these suggestions, and I'd like to encourage the authors to make the corresponding changes.

Reviewer's Responses to Questions

**Comments to the Authors:**

Reviewer #1: This paper describes a carefully thought out model of collective foraging in social insect colonies, and shows a new potential role for negative signals within collective foraging processes. While such signals had previously been assumed to be beneficial due to their role in directly communicating a negative state (lack of food, danger etc), this model tests the idea that such signals could also make forager allocation more robust, especially in limited foraging populations, as social insects frequently have.

The model description is very well laid out. The logic underlying the assumptions and decisions is clearly explained. The supplementary material is likewise clear, and support the main results well.

The duel modelling approaches are a particular strength of the paper, and the differences in the outcomes between the two modelling approaches are clearlydemonstrated and explained.

I did find table 1 a bit confusing, because at first reading, it seemed to suggest that the negative-feedback model also has an additional change to the positive feedback process, thus confounding the effect due to negative feedback. After careful reading of the text, I see that this is not the case, but the terminology is a bit non-intuitive. It would be possible to have quality-dependent positive-only recruitment, and I think that some small changes to the section of the text that justifies the introduction of quality dependence (lines 87-90) and/or to table 1 could make things clearer in this regard.

To summarise, I have no substantive concerns about this manuscript, and consider it to provide novel insights to the field of collective decision making.

Reviewer #2: This manuscript provides an interesting and important new potential function for inhibitory signals in social insect collectives. By using the example of the honey bee stop signal, it shows, with modeling, that adding negative feedback allows colonies with a realistic number of foragers to arrive at the correct foraging distributions. This is interesting because assuming an infinite colony size results in correct distributions, with and without negative social feedback, but at realistic colony sizes, the colony can arrive at decisions that are not optimal, based upon stochastic initial conditions. Further, errors in achieving the correct distribution are significantly higher without such negative feedback.

I think the manuscript is well written and the results clearly explained. Although I cannot comment on the details of the modeling, the results make sense given what is known about how collectives behave in response to such negative feedback.

One suggestion is that the authors consider the recent paper by Bell et al. (2021) showing that response to stop signals changes as a function of colony size and incorporate this in their discussion. It would be interesting to consider this paper in light of Fig. 3 given that the response of colonies of different sizes are, in a way, similar to considering the different stop signaling strength levels shown in this figure.

Bell, Heather C., et al. "Responsiveness to inhibitory signals changes as a function of colony size in honeybees (Apis mellifera)." Journal of the Royal Society Interface 18.184 (2021): 20210570.

With respect to Fig. 3, I would welcome more discussion or explanation. It seems that the optimal stop signal strength occurs at z=0.4 but that increasing z results in a very large increase in predicted error. Is this a result of using the infinite population model? Would using a finite, more realistic population model yield lower predicted errors as z increases? Also, the inset figure with recruitment strength on the x-axis should be mentioned in this legend.

In terms of the text, I have a small suggestion:

L155. I suggest, “As a motivating example, decreases in the number of waggle circuit repetitions in honeybee swarms” to clarify that the waggle phase durations are not decreasing, jut the number of waggle dance repetitions.

Reviewer #3: The authors identify an interesting and potentially generalizable finding about the role of negative feedback in collective decisions. Namely, they have developed a stochastic dynamical model of collective foraging that involves the key behaviors of discovery, abandonment, recruitment, and stop-signaling (the primary negative feedback interaction). The employ both a mean field model and a fully stochastic model to examine the relative contribution of negative feedback to the speed of approach to and relative value of the quasiequilibria of the stochastic system. Both a model without and a model with negative feedback exhibit steady states aligned or roughly aligned to an ideal free distribution in the mean field limit. However, when finite size effects are considered, the model without negative feedback does not reliably approach the desired distribution. Lastly, it is found that mild deviations in the eventual distribution increase for stronger negative feedback, so there is an optimum level of feedback that best provides reduced convergence time and relatively low error.

The paper is short, but the main findings are apparent in the results section and figures. What is not exactly clear to me is how the model itself brings about the reduction in variance, even in the supplemental information. Perhaps this is primarily a simulation-based result, but it seems like the authors should be able to explain in more detail precisely how the variance-reduction mechanism works. My intuition would tell me that perhaps the additional term is somehow increasing the stability (increasing the amplitude of the main negative eigenvalue) associated with the steady state. Is this something the authors could show for the full stochastic system? Is this something that would appear in a system-size expansion approximation of the full stochastic system? Seems like there should be some analysis that could be done to make it clearer. I would appreciate any response the authors could give to these queries in a revision.

Other than this, I would say that the authors could probably move some of the results from the Supplementary Info document up to the main text. It's written rather like a PRL or PNAS article with a lot of interesting analysis tucked into the supplement, but I'd say the sections ST1 and ST3 and their figures could be moved up to the main text.

The model is not particularly novel by any means and it certainly has been analyzed by these authors and others a lot in previous works, but nevertheless this study sheds light on a newly observed dynamical phenomenon of the model. The authors properly frame the findings and provide a nice set of analyses. As mentioned above, the paper would primarily benefit from a clearer and more detailed explanation of exactly (in terms of a dynamical systems or stochastic processes analysis if possible) how negative feedback brings about variance reduction. If the authors could add more detail on this, I would be happy to consider a revision.

**Have the authors made all data and (if applicable) computational code underlying the findings in their manuscript fully available?**

Reviewer #1: Yes

Reviewer #2: Yes

Reviewer #3: Yes

PLOS authors have the option to publish the peer review history of their article (what does this mean?). If published, this will include your full peer review and any attached files.

Reviewer #1: **Yes: **Elva JH Robinson

Reviewer #2: No

Reviewer #3: No

Figure Files:

Data Requirements:

Reproducibility:

References:

---

## [Decision Letter · Decision Letter 1]

8 Apr 2022

Dear Dr Reina,

We are pleased to inform you that your manuscript 'Negative feedback may suppress variation to improve collective foraging performance' has been provisionally accepted for publication in PLOS Computational Biology.

Best regards,

Christian Hilbe

Associate Editor

PLOS Computational Biology

Natalia Komarova

Deputy Editor

PLOS Computational Biology

The manuscript has been sent to two of the reviewers who handled the original manuscript.

Both reviewers are happy with the changes, and so am I.

(I am personally leaning with reviewer #3 about moving more of the supplementary material into the main text; however, if the authors prefer to keep their manuscript as is, that's fine for me).

Reviewer's Responses to Questions

**Comments to the Authors:**

Reviewer #2: The authors have addressed my comments and I believe that the manuscript is now ready for publication.

Reviewer #3: Thanks to the authors for considering my comments and pursuing additional analysis to test the origins of the variance-reduction mechanism. The response makes good sense, and I think the resulting update to the paper is nice. Concerning moving supplementary up to the main text, I guess that's my bias as an applied mathematician, and I could see that for biologists this could be too much. The paper in its current form is in good shape for publication in PLoS Comput. Biol. as far as I'm concerned.

**Have the authors made all data and (if applicable) computational code underlying the findings in their manuscript fully available?**

Reviewer #2: Yes

Reviewer #3: Yes

PLOS authors have the option to publish the peer review history of their article (what does this mean?). If published, this will include your full peer review and any attached files.

Reviewer #2: No

Reviewer #3: No

---

## [Editor Report · Acceptance letter]

11 May 2022

PCOMPBIOL-D-22-00055R1 

Negative feedback may suppress variation to improve collective foraging performance

Dear Dr Reina,

I am pleased to inform you that your manuscript has been formally accepted for publication in PLOS Computational Biology. Your manuscript is now with our production department and you will be notified of the publication date in due course.

With kind regards,

Livia Horvath
